# Mapping the Interactome of KRAS and Its G12C/D/V Mutants by Integrating TurboID Proximity Labeling with Quantitative Proteomics

**DOI:** 10.3390/biology14050477

**Published:** 2025-04-26

**Authors:** Jiangwei Song, Busong Wang, Mingjie Zou, Haiyuan Zhou, Yibing Ding, Wei Ren, Lei Fang, Jingzi Zhang

**Affiliations:** 1Department of Oncology, Nanjing Drum Tower Hospital, Affiliated Hospital of Medical School, Nanjing University, Nanjing 210093, China; 502022350034@smail.nju.edu.cn (J.S.); zhangjingzi@nju.edu.cn (J.Z.); 2State Key Laboratory of Pharmaceutical Biotechnology, Jiangsu Key Laboratory of Molecular Medicine, Chemistry and Biomedicine Innovation Center, Medical School of Nanjing University, Nanjing 210029, China; 602023350041@smail.nju.edu.cn (B.W.); 502023350059@smail.nju.edu.cn (M.Z.); 502023350056@smail.nju.edu.cn (H.Z.); missdyb@nju.edu.cn (Y.D.)

**Keywords:** KRAS interactome, KRAS G12 mutants, TurboID proximity-labeling, quantitative proteomics, metabolic reprogramming

## Abstract

KRAS gene mutations are key drivers of many cancers, but how these mutations alter cellular processes to promote tumor growth remains unclear. This study investigates how specific KRAS mutations disrupt protein interactions to rewire cancer cell metabolism. We developed a novel approach combining protein labeling and advanced analysis to compare the interaction partners of normal KRAS with three common cancer-causing mutants (G12C, G12D, and G12V). Our findings reveal that these mutations dramatically change KRAS’s ability to bind proteins controlling insulin signaling, energy production, and stress responses. These altered interactions force cells into abnormal metabolic states that accelerate tumor growth and help evade immune defenses. Notably, all three mutations weaken connections with a protein responsible for breaking down KRAS while strengthening ties to another protein that drives cell growth. By mapping these mutation-specific changes, we explain how different KRAS variants promote cancer and identify new targets for precision therapies. This work provides crucial insights for developing treatments that specifically counteract the effects of particular KRAS mutations, offering hope for more effective anti-cancer strategies.

## 1. Introduction

The Kirsten rat sarcoma viral oncogene homolog (KRAS), as the most prevalent subtype of the RAS proto-oncogene family—the earliest discovered in humans—exhibits mutations in approximately 20% of malignancies with distinct tissue-specific distributions [1]. KRAS mutations occur most frequently in pancreatic adenocarcinoma (PAAD, ~90%) [2], followed by colorectal cancer (CRC, 36–40%) [3], cholangiocarcinoma (CHOL, ~27%) [4], and lung adenocarcinoma (LUAD, 15–20%) [5]. These mutations predominantly consist of single-base missense substitutions, leading to the constitutive activation of KRAS. Over 81% of these substitutions occur at codon 12 (G12), with common variants including G12C, G12D, and G12V [6,7,8,9]. In PAAD, G12D accounts for 45% of G12 mutations, followed by G12V (35%) and G12R (17%), whereas G12C is rare (2%). Conversely, G12C is the predominant mutation in LUAD (46%) [10,11,12]. These mutations not only serve as biomarkers for disease staging but also significantly impact the overall survival of patients during clinical cancer therapy [13].

Due to its high mutation frequency, overexpression, and persistent activation in cancers, KRAS has long been regarded as the “holy grail” for targeted anticancer drug development. However, the therapeutic targeting of KRAS remains challenging owing to its molecular characteristics: a compact globular structure (189 amino acids; ~21 kDa molecular weight) [14], a smooth protein surface lacking deep hydrophobic pockets for small-molecule binding [15], and picomolar affinity for guanosine triphosphate/guanosine diphosphate (GTP/GDP) in the presence of abundant intracellular nucleotides, which creates pharmacological competition [16,17]. Collectively, these features rendered KRAS an “undruggable” target in oncology for decades [5].

KRAS activation requires upstream EGFR signaling [18] and multiple post-translational modifications, including C-terminal cysteine prenylation for membrane localization [19]. Once activated, KRAS participates in subsequent signal transduction to regulate phosphatidylinositol 3-kinase–protein kinase B (PI3K-AKT) and mitogen-activated protein kinase (MAPK) signaling pathways [20]. Oncogenic mutations lock KRAS in a GTP-bound state, resulting in sustained pathway activation that drives uncontrolled proliferation and immune evasion—central mechanisms of carcinogenesis. Therefore, KRAS mutations have already emerged as a major driver of carcinogenesis.

Current therapeutic strategies targeting KRAS mutations focus on three approaches: (1) direct inhibition of mutant KRAS, (2) blockade of membrane localization, and (3) suppression of downstream signaling. However, the complex molecular mechanisms of KRAS-driven oncogenesis and its central role in cellular signaling networks have limited clinical progress. Most targeted inhibitors remain in early-stage trials, and acquired resistance frequently develops in patients [21].

In this study, we integrated the TurboID proximity labeling (PL) technique with quantitative proteomics to characterize the interacting proteins of the KRAS wild type (WT) and its G12C/D/V mutants and comprehensively generated the protein–protein interaction (PPI) networks of each KRAS. Further bioinformatics analyses revealed KRAS G12C/D/V mutant-dependent changes in KRAS binding proteins and their enrichment pathways, particularly in glucose and lipid metabolism, insulin signaling pathways, and reactive-oxygen-species-related pathways. Notably, all KRAS G12C/D/V mutants showed similar interactome shifts due to reduced leucine-zipper-like transcription regulator 1 (LZTR1) binding, enhanced late endosomal/lysosomal adaptors, and MAPK and mTOR activator 1 (LAMTOR1) recruitment. By analyzing shared/differential metabolic pathway changes across different KRAS G12C/D/V mutants, we have established a proteomic framework that not only explains KRAS mutation-driven tumorigenesis but also identifies KRAS mutation-specific metabolic reprogramming for new therapeutic target exploitation.

## 2. Materials and Methods

### 2.1. Construction of KRAS Wild-Type and G12 C/D/V Mutant Stable Cell Lines

#### 2.1.1. Plasmid Construction and Sequence Verification

We engineered an affinity-purified HA tag tandem TurboID for the N-terminus of the KRAS via PCR homologous recombination. The recombinant fragments were cloned into the pLVX lentiviral vector using a BamHI (*Sangon*, Shanghai, China) single-restriction enzyme cloning strategy (restriction site selection guided by KRAS conserved domains in GenBank records). All plasmid constructs underwent Sanger sequencing (*Genewiz*, Suzhou, China) followed by sequence alignment against the NCBI KRAS reference (NM_004985.4) using *SnapGene* v6.0.2. Verified HA-TurboID-KRAS WT plasmids were transformed into Stbl3 chemically competent cells (*TransGen Biotech*, Beijing, China) for large-scale propagation. Primer design involved four specific sequences (*Genewiz*, Suzhou, China) (Table 1).

To generate KRAS G12C/D/V mutants, we performed site-directed mutagenesis on WT plasmids using the QuickMutation™ Kit (D0206S, *Beyotime Biotechnology*, Shanghai, China) with eight customized primers (Table 2).

Nucleotide substitutions were introduced to generate glycine-to-valine/cysteine/aspartic acid mutations at codon 12th, corresponding to G12C (GGT→TGT), G12D (GGT→GAT), and G12V (GGT→GTT) variants.

#### 2.1.2. Plasmid Transformation and Amplification

After plasmid construction, heat-shock transformation was performed using Stbl3 competent cells. Transformed colonies were cultured in a Luria–Bertani (LB) liquid medium with Ampicillin (working concentration 1:1000) under aerobic conditions (37 °C) for 14–16 h to achieve high-copy plasmid amplification. Bacterial pellets were harvested by centrifugation (4000 rpm, 10 min, 4 °C) prior to plasmid extraction using the HiPure Plasmid Miniprep Kit (Cat# DC201-01, *Vazyme Biotechnology*, Nanjing, China). Purified plasmids were eluted in 50 μL Elution Buffer and cryopreserved at −20 °C for later applications.

#### 2.1.3. Lentivirus Infection and Biological Activity Verification

HEK293T cells were utilized for lentivirus production. Cells were seeded in 10 cm dishes and cultured with fresh Dulbecco’s modified eagle medium (DMEM, Cat# 319-075-CL, *Wisent*, Nanjing, China) supplemented with 10% fetal bovine serum (FBS, Cat# FSD500, *ExCell Bio*, Shanghai, China) and 1% streptomycin/penicillin (Cat# C100C5, *CELLSAVING*, Suzhou, China). For lentiviral packaging, a plasmid mixture containing the KRAS expression vector (or tag-only plasmid as blank control), psPAX2, and pMD2.0G was prepared at a 3:2:1 mass ratio, using triple plasmid volume polyethylenimine (PEI, *Solarbio*, Shanghai, China) as the co-transfection reagent. Viral supernatants were collected 48 h post-transfection and filtered through a 0.45 μm membrane. For infection, HEK293T cells seeded in 6-well plates were treated with viral supernatants containing 8 μg/mL Polybrene as a co-infection reagent. Stable cell lines were selected using 2 μg/mL Puromycin for 72 h following two rounds of infection. Surviving cells expressing HA-TurboID (Tag-only, blank control) and HA-TurboID-KRAS (WT, G12C, G12D, and G12V) were expanded for subsequent experiments. The Western blot analysis of whole-cell lysates confirmed the stable overexpression of HA-TurboID and HA-TurboID-KRAS (WT, G12C, G12D, and G12V). The antibodies used were HA (Cat# AE105, *Abclonal*, Wuhan, China) and β-actin (Cat# AC026, *Abclonal*, Wuhan, China). To verify biological activity, Cell Counting Kit-8 (CCK-8, Cat# CK04, *Dojindo*, Kumamoto, Japan) assays were performed on five experimental groups: blank control, WT, G12C, G12D, and G12V. Each group included six technical replicates.

### 2.2. TurboID Proximity Labeling for Enrichment of KRAS Interacting Proteins

Stable cell lines obtained from previous experiments were expanded to a density of 1.6 × 10⁸ cells. Biotin (50 μM) was introduced to the culture medium for a 2 h incubation period (blank group only incubated for 0.5 h), enabling the biotinylation of KRAS proximal proteins through TurboID biotin ligase activity. Biotinylated proteins were affinity-purified using NeutrAvidin™ agarose resin (Cat# 31000, *Thermo Fisher Scientific*, Waltham, Massachusetts, USA) through 14 h rotation at 4 °C.

Whole-cell lysates (input) and NeutrAvidin-enriched proteins (AP) were then analyzed by Western blot. The antibodies used were HA and HRP-conjugated Streptavidin (Cat# SA00001-0, *Proteintech*, Wuhan, China).

About protein extraction and Western blot, first, total proteins from cultured cells were extracted in a RIPA buffer supplemented with PSMF (0.1 mM) for 10 min on ice and then centrifuged at 10,000× *g* for 10 min. The protein concentration was measured using a BCA protein assay kit (Cat# 23225, *Thermo Fisher Scientific*, Waltham, Massachusetts, USA). Protein samples were subjected to electrophoretic separation on SDS-PAGE and transferred to the PVDF membrane. The membrane was blocked with 5% milk in Tris-buffered saline with 0.1% Tween-20 for 2 h and then incubated with primary antibodies overnight and secondary antibodies for 1 h at RT. The antibody used against the LZTR1 (Cat# YN1668) protein was purchased from *Immunoway* (Texax, CA, USA). Antibodies used against LAMTOR1 (C11orf59, Cat# F0691) proteins were purchased from *Selleck* (Shanghai, China). The antibody used against the β-actin (Cat# 2D4H5) protein was purchased from *Proteintech* (Wuhan, China). All primary antibodies were used at 1:1,000. HRP-conjugated Affinipure Goat anti-Rabbit IgG (Cat# SA00001-2, *Proteintech*, Wuhan, China) and HRP-conjugated Affinipure Goat anti-Mouse IgG (Cat# SA00001-1, *Proteintech*, Wuhan, China) were used at 1:5000. Band intensities were quantified using *Image J* (*National Institutes of Health*, Maryland, USA).

### 2.3. Quantitative Proteomic Characterization of KRAS Interacting Proteins

#### 2.3.1. Sample Preparation

Enriched proteins were resolved by 10% SDS-PAGE (Cat# PG112, *Epizyme*, Shanghai, China) and visualized with Coomassie brilliant blue R-250 staining. Each protein lane was excised into 1 mm^3^ gel fragments for subsequent MS sample pretreatment. Destaining was performed with 25 mM NH_4_HCO_3_ containing 5% acetonitrile (ACN). The reduction in disulfide bonds was achieved using 10 mM dithiothreitol (DTT) at 56 °C for 45 min, followed by alkylation with 20 mM iodoacetamide (IAM) in the dark for 45 min.

Trypsin digestion (1:100 ratio in 25 mM NH_4_HCO_3_) was conducted at 37 °C for 14–16 H. The digestion was terminated with 10% formic acid (FA), and peptides were extracted using 50% ACN/5% FA. Peptide solutions were concentrated to dryness using a CentriVap (*LABCONCO,* Kansas, MO, USA) benchtop centrifugal concentrator and reconstituted in 3% ACN/2% FA prior to LC-MS/MS analysis.

#### 2.3.2. LC-MS/MS Analysis

Samples were analyzed using an ACQUITY UPLC M-Class system (*Waters*, Milford, Massachusetts, USA) coupled to a ZenoTOF 7600 mass spectrometer (*SCIEX*, Boston, Massachusetts, USA). The mass spectrometer adopts Electron Activation Dissociation (EAD) and Zeno™ Trap technology to enhance sensitivity. The chromatographic separation buffer was a 0.1% FA /2% ACN water solution (Buffer A) and 0.1% FA/2% water acetonitrile solution (Buffer B). The mobile phase is separated by a linear gradient at a 5 μL/min flow rate. The peptide resolution was loaded into the chromatographic injection bottle (Cat# VDAP-4025PBS-631-100, *CNW*, Shanghai, China), and a 4 μL sample volume was used per injection. MS data were acquired in the data-dependent acquisition (DDA) mode. The reagents used for sample pretreatment and loading were of mass spectrometry grade and purchased from *Sigma-Aldrich* (St. Louis, MO, USA).

#### 2.3.3. Data Processing and Bioinformatics Analysis

The original mass spectrometry data were analyzed by *ProteinPilot* 5.0.2 (*SCIEX*, Boston, MA, USA) software, and the experimental MS/MS secondary spectrum was matched with the theoretical spectrum through the NCBI database to obtain reliable peptide information preliminarily. Subsequently, a secondary search of the peptide information library was conducted using the *ProteinProspector* v6.6.6 platform (https://prospector.ucsf.edu/prospector/mshome.htm. Accessed on 6 December 2024) with the *SwissProt* database (release 18 June 2021; *Homo sapiens*). This two-stage analysis yielded fundamental protein characteristics, including the protein’s accession, gene name, unique number, peptide count, and protein name.

The functional annotation of the KRAS wild type’s and mutant’s interacting partners was performed through Gene Ontology (GO) classification and Kyoto Encyclopedia of Genes and Genomes (KEGG) pathway analysis. Protein–protein interaction (PPI) networks were concurrently constructed by *STRING* (string-db.org/. Accessed on 6 March 2025). Data visualization was achieved using R package *ggplot2* (v3.5.1) for bubble plots and *Venn Diagram* (v1.7.3) for setting analysis. Finally, target protein expression patterns in malignancy contexts were validated through differential expression profiling against the TCGA repository (https://cancergenome.nih.gov/. Accessed on 10 February 2025).

### 2.4. Statistical Analysis

All statistical analyses were performed using *GraphPad Prism* 8.0.1, and data are expressed as mean ± SD from three independent experiments (*n* = 3). Statistical analyses were performed using a two-tailed unpaired Student’s *t*-test for a two-group comparison or a one-way ANOVA followed by multiple comparisons using the LSD post hoc test for more than two groups. * *p* < 0.05, ** *p* < 0.01, and *** *p* < 0.001, and n.s. means no significance.

## 3. Results

### 3.1. An Integrated Strategy of Mapping KRAS-Interacting Proteins

In this study, we utilized TurboID proximity labeling to systematically identify interactors of KRAS-WT and its three common KRAS G12C/D/V mutants. The experimental workflow was initiated with the genetic engineering of KRAS constructs (Figure 1). Briefly, we engineered the KRAS-WT/Mutant plasmids through sequential modifications: site-specific insertion of an HA-TurboID tandem tag followed by codon 12 mutations (KRAS G12C/D/V). Using lentiviral transduction, we established HEK293T cell lines constitutively expressing HA-TurboID-KRAS variants (WT, KRAS G12C/D/V), with puromycin selection ensuring stable transgene expression while minimizing variability from transient transfection. The experimental design included four KRAS variant groups (HA-TurboID-KRAS fusion proteins) and one control group (HA-TurboID only). Following biotin treatment to activate TurboID-mediated proximity labeling, KRAS-neighboring proteins were covalently biotinylated, captured via NeutrAvidin bead purification, and analyzed by quantitative mass spectrometry. This approach successfully resolved distinct interactomes between KRAS WT and G12C/D/V mutants, concurrently revealing their characteristic metabolic pathway alterations (Figure 1).

HA-TurboID-KRAS consists of a total of 1554 bases, with mutation codons 1042 to 1044. The sequence integrity of the KRAS WT and G12C/D/V mutants’ plasmids was confirmed by DNA sequencing (Figure 2A). Western blot confirmed the comparable expression levels of HA-TurboID-KRAS fusion proteins across all stable cell lines (Figure 2B). CCK-8 assays indicated accelerated proliferation rates in KRAS G12C/D/V mutant-expressing cells relative to WT controls (Figure 2C). Streptavidin blot analysis demonstrated enhanced biotinylation signals in KRAS-expressing groups compared to HA-TurboID tag-only controls (Figure 2D, Appendix A), confirming the efficient labeling of interactors.

Applying stringent selection criteria (peptide count (PC) ≥ 2; fold change (FC) ≥ 2 enrichment vs. control), we defined the proteins identified only in the experimental group as KRAS-specific binding proteins and the proteins with FC ≥ 2 as strongly KRAS -interacting proteins; these two components make up the KRAS proximal binding protein. Under these criteria, screening obtained 174 proteins in the WT group (127 specific interactors; 47 proteins with FC ≥ 2); 131 proteins in G12C (100 specific; 31 proteins with FC ≥ 2); 219 proteins in G12D (170 specific; 49 proteins with FC ≥ 2); and 94 proteins in G12V (74 specific; 20 proteins with FC ≥ 2). An UpSet plot was obtained to visualize both shared and variant-specific interaction networks among the four groups (Figure 2E). Subsequent bioinformatics analysis was conducted to reveal KRAS-mutant-specific metabolic reprogramming pathways.

### 3.2. KRAS G12C/D/V Mutations Induce Intracellular Metabolic Reprogramming

#### 3.2.1. Pathway and Functional Enrichment Analysis of KRAS WT-Interacting Proteins

To characterize the KRAS interactome in the absence of mutation, we performed KEGG and GO enrichment analyses on the 174 specific binding proteins of KRAS WT. KEGG pathway analysis revealed that KRAS and its proximal interacting proteins were significantly enriched in the MAPK signaling pathway (hsa04810) and the PI3K-AKT signaling pathway (hsa04914) (Figure 3A). KRAS WT interactome analysis revealed moderate enrichment in the MAPK (*p* = 0.047) and PI3K-AKT (*p* = 0.032) pathways—canonical KRAS effectors driving proliferation in PAAD and CRC. Notably, mutant-specific pathways such as autophagy (*p* = 0.002) and cell adhesion (*p* = 0.001) exhibited stronger statistical significance, reflecting emerging metabolic vulnerabilities critical for KRAS-mutant cell survival under stress (Figure 3A). These related metabolic pathways involve the classical downstream effectors of KRAS, including RAF1, ARAF, BRAF, and EGFR, which are critical for regulating cell proliferation, differentiation, and inflammatory responses. Biological process (BP) and molecular function (MF) analyses show that KRAS interactors bind to intercellular adhesion molecules and small G proteins, regulating KRAS target membrane localization processes and the cell–cell junction (Figure 3B). These findings suggest that KRAS is positioned at the core of the intracellular signal transduction network, exerting its regulatory effects through its GTPase activity and multiple binding sites for downstream effectors [22]. These properties enable KRAS to play a central role in various cellular processes, including signal transduction, proliferation, and inflammation [23,24].

#### 3.2.2. Dynamic Changes in KRAS Interacting Proteins Caused by KRAS G12C/D/V Mutations

The KRAS G12C mutation group exhibited 131 binding proteins. Pathway enrichment analysis revealed significant differences compared to the WT group in pathways such as leukocyte transendothelial migration (hsa04670), chemical carcinogenesis-reactive oxygen species (hsa05208), and cellular senescence (hsa04218) (Figure 3C). These pathways are related to immune regulation.

Similarly, the KEGG analysis of the 219 binding proteins in the KRAS G12D group revealed significant enrichment in metabolic pathways, including the insulin signaling pathway (hsa04910), mTOR signaling pathway (hsa04150), and fatty acid metabolism and fatty acid biosynthesis (hsa00061, hsa01212) (Figure 3D). These signaling pathways control intracellular glucose and lipid metabolism and provide the energy foundation for cell proliferation. Notably, the KRAS G12D oncogenic mutation also induces the reprogramming of intracellular glucose, lipid, and amino acid metabolism pathways, including N-sugar biosynthesis, the pentose phosphate pathway (PPP), fatty acid metabolism and biosynthesis, and amino acid biosynthesis (Figure 3D). Compared to normal cells, KRAS G12D mutant cells favor the pentose phosphate pathway (PPP) for glucose metabolism even under aerobic conditions. This increases glucose demand to support rapid cell proliferation. Abnormal lipid metabolism is another feature of metabolic remodeling in KRAS G12D mutant cells. Pathway enrichment analysis revealed the significant enrichment of fatty acid synthesis-related pathways (hsa00061). Proteins such as fatty acid synthase (FASN) and acetyl-CoA carboxylase alpha (ACCA) are highly enriched in KRAS G12D mutation interactors, driving intracellular lipid anabolism to meet energy demands during rapid cell growth. The amino acid metabolic pathway also responded to the KRAS G12D mutation. Multiple amino acid transporters, including SLC1A5, SLC38A1, and SLC7A5, are enriched in the KRAS G12D mutation interactome. These transporters facilitate essential amino acid glutamine (Gln) uptake in tumor cells. Additionally, KRAS G12D mutation regulates the transcription of key metabolic enzymes such as glutamate dehydrogenase (GLUD1) and aspartate aminotransferase (GOT1) [25,26], enabling tumor cells to meet carbon and nitrogen demands for rapid proliferation and macromolecular synthesis.

The KRAS G12V mutation group exhibited 94 binding proteins. Pathway enrichment analysis revealed significant enrichment in the insulin signaling pathway (hsa04910) and fatty acid metabolism and fatty acid biosynthesis (hsa00061, hsa01212) (Figure 3E). Thus, similarly to the KRAS G12D mutation, the KRAS G12V mutation probably affects energy metabolism pathways such as glucose and lipid metabolism, contributing to the malignant transformation of cells.

In addition to KEGG analysis, GO enrichment annotation was also conducted on the proximal binding proteins of three mutant groups, which are integral to shared BP such as plasma membrane localization and the regulation of small G protein family signal transduction. These mutant groups exhibited analogous MF, including small GTPase binding and cadherin binding, as well as the formation of comparable cellular components (CC) (Figure A1). Notably, proteins within the G12C group are integral to maintaining the blood–brain barrier and tissue homeostasis. These proteins possess cyclin-dependent protein serine/threonine kinase activity and are capable of activating a variety of effectors. Conversely, proteins in the G12D group are associated with immune response-activating cell surface receptor signaling pathways and exhibit active transmembrane transporter activity. Furthermore, in the G12V group, some proteins have phosphatidylinositol 3-kinase (PI3K) binding functions, including PTPN13, IRS4, and ATP1A1, which regulate the PI3K-AKT signaling pathway.

### 3.3. Simultaneously Changed Interacting Proteins in All KRAS G12C/D/V Mutants

#### 3.3.1. Simultaneously Decreased Interacting Proteins in All KRAS G12C/D/V Mutants

To investigate the effects of KRAS G12C/D/V mutants on KRAS interactors, we analyzed the interacting proteins that simultaneously changed in KRAS G12C/D/V mutants, using the WT as a control. Proteins with an FC ≥ 2 in the mutant group were classified as increased-binding proteins, while those with FC ≤ 0.5 were classified as decreased-binding proteins. To ensure data reliability, we applied a quality control criterion requiring the peptide count to be ≥ 2. Through this screening process, we identified 100 proteins with reduced binding in the KRAS G12C mutation group, 72 proteins in the KRAS G12D mutation group, and 192 proteins in the KRAS G12V mutation group (Figure 4A, Appendix A).

To further characterize these decreased proteins, we performed a Venn diagram analysis and identified 45 proteins with reduced binding in all three mutation groups (Figure 4B, Appendix A). These proteins were defined as potential negative regulatory factors for the malignant transformation of KRAS G12C/D/V mutants’ cells. To elucidate their biological roles, we conducted KEGG and GO pathway analyses. The KEGG analysis revealed the significant enrichment of these proteins in the following pathways: Nucleocytoplasmic transport (hsa03013), protein processing in the endoplasmic reticulum (hsa04141), citrate cycle (TCA cycle) (hsa00020), and endocytosis (hsa04144) (Figure 4C). KRAS G12C/D/V mutations can diminish the binding interaction with nucleoporin (NUP160/107) and transcription factor chaperones YWHAB (14-3-3β). The dysregulation of this pathway can disrupt the expression of transcription factors and nucleocytoplasmic transport machinery, thereby disturbing the balance between cell proliferation and apoptosis. Additionally, EGFR, EPS15L1, and AGAP3, which are linked to endocytosis, showed altered interactions. The dysregulation of this pathway may promote malignant cell proliferation and immune evasion. GO enrichment annotation revealed that the co-decreased binding protein LZTR1 is associated with the CUL3-RING E3 ubiquitin ligase complex (CRL3) and involved in the regulation of small GTPase-mediated signal transduction (Figure 4D). As an adaptor protein for the E3 ubiquitin ligase Cullin 3 (CUL3), LZTR1 forms a complex with CUL3 to ubiquitinate KRAS at a specific amino acid (K170) [27,28], thereby disrupting its membrane localization. The proteomics data indicate that the binding of KRAS to LZTR1 is reduced in KRAS mutations. This disruption affects the ubiquitination and degradation of KRAS *via* the LZTR1-CUL3 axis. Therefore, LZTR1 could be a potential regulatory mechanism for KRAS-activating mutations.

Additionally, we conducted PPI analysis using STRING for the co-decreased binding proteins in the KRAS G12C/D/V mutant groups. The analysis revealed that KRAS is involved in a protein cluster described as Tristetraprolin (TTP and ZFP36) and binds and destabilizes mRNA (Figure 4E).

#### 3.3.2. Simultaneously Increased Interacting Proteins in All KRAS G12C/D/V Mutants

Based on the established screening criteria, 64 proteins were identified with increased binding in the KRAS G12C group, 157 proteins in the KRAS G12D group, and 39 proteins in the KRAS G12V group (Figure 4B). To further analyze the overlap of proteins with increased binding across the three groups, a Venn diagram was employed. This analysis revealed that four proteins exhibited increased binding in all three mutation groups (Figure 4B). Among them, LAMTOR1 plays an important role in intracellular metabolism. As a key component of the Ragulator pentamer, LAMTOR1 plays a critical role in targeting the lysosomal membrane for localization and facilitating the binding of the Ragulator complex to Rag guanosine triphosphate through its self-ubiquitination activity [29]. This, in turn, activates the mammalian target of rapamycin 1 (mTORC1) signaling pathways [25]. Notably, studies have demonstrated that inhibiting the ubiquitination of LAMTOR1 suppresses mTORC1 activation [30], thereby accelerating inflammatory responses. According to data from the TCGA database, LAMTOR1 is highly expressed in various malignant tumors, such as hepatocellular carcinoma (HCC) and colon adenocarcinoma (COAD) [31]. As a protein with increased binding in KRAS G12C/D/V mutation groups, LAMTOR1 may emerge as a potential therapeutic target for cancer treatment.

#### 3.3.3. Validation of Simultaneously Changed Interacting Proteins in KRAS G12C/D/V Mutants

Based on the mass spectrometry data, two potential target proteins, LZTR1 and LAMTOR1, were identified as the co-decreased and co-increased binding proteins in KRAS G12C/D/V mutants for biochemical validation. Western blotting validation was conducted to confirm their differential binding to KRAS WT and G12C/D/V mutants (Figure 4F–J, Appendix A). The results showed that both proteins exhibited comparable expression levels in the whole-cell lysates of the control, KRAS WT, and G12C/D/V mutants’ stable cell lines. However, in the KRAS-binding proteins enriched *via* NeutrAvidin affinity purification, LAMTOR1 demonstrated notably higher expression, whereas LZTR1 exhibited reduced expression in the KRAS G12C/D/V mutant groups compared to the WT group. While baseline LAMTOR1 expression varied across groups (Input lanes), NeutrAvidin enrichment values reflect mutant-specific binding efficiencies rather than expression levels. These data collectively support the fact that KRAS G12C/D/V mutations significantly affect the KRAS-binding ability with their important interactors.

## 4. Discussion

KRAS mutations are now recognized as pivotal drivers across multiple cancer types, with codon 12 (G12) mutations—particularly G12C, G12D, and G12V—representing the most frequent oncogenic variants [32]. While tissue-specific differences in mutation prevalence and carcinogenic mechanisms exist [33,34], all KRAS mutants converge on aberrant proliferation and malignant transformation through dysregulated signaling. Our study advances this understanding by systematically mapping mutation-specific interactomes and their metabolic consequences, offering actionable insights for therapeutic development.

By constructing HA-TurboID-tagged KRAS WT and G12C/D/V mutant plasmids, we established stable HEK293T cell lines to profile proximal interactors. TurboID’s biotin ligase activity enabled the efficient capture of transient or weak interactions, outperforming BioID in labeling efficiency and stability [35,36]. Unlike APEX2, TurboID operates without cytotoxic H_2_O_2_ induction [37], making it ideal for dynamic interaction studies.

Pathway enrichment analysis demonstrated that interacting proteins across all KRAS variants (WT, G12C, G12D, and G12V) were enriched in classical KRAS signaling pathways, including PI3K/AKT and MAPK pathways. Consistent with this, canonical KRAS interactors such as EGFR and RAF [38] were identified in all groups, confirming that KRAS G12 mutations preserve the core signaling functions of KRAS. However, each mutant induced distinct metabolic perturbations. The KRAS G12C mutation activated leukocyte transendothelial migration (TEM), chemical carcinogenesis-related reactive oxygen species (ROS), and cellular senescence pathways. These findings suggest that the KRAS G12C mutant promotes CD45^+^ leukocyte infiltration via TEM signaling [39], triggering ROS and interleukin secretion by immune cells. While this inflammatory response may enhance tumor surveillance [40], its dual role in tumor promotion and suppression warrants further investigation. In contrast, the KRAS G12D mutant upregulated insulin signaling, fatty acid synthesis, and the pentose phosphate pathway. Elevated insulin levels in KRAS-mutant pancreatic acinar cells drive trypsinogen autoactivation and secretory stress, accelerating pancreatic intraepithelial neoplasia (PanIN)—a key precursor to PAAD [41]. Similarly, the KRAS G12V mutation shared metabolic features with KRAS G12D but exhibited stronger coupling to central carbon metabolism, promoting aerobic glycolysis and lipid anabolism to fuel proliferation. Notably, all mutants maintained GTP-bound activation [42], perpetuating downstream proliferative signals. Paradoxically, mutant KRAS also stimulated immune cell infiltration, which might transiently counteract malignant progression.

The comparative analysis against KRAS WT revealed 45 proteins with consistently reduced binding across mutants. Most strikingly, the interaction with LZTR1—a CUL3 E3 ligase adaptor critical for KRAS ubiquitination and degradation—was diminished in all mutants. This defect likely stabilizes KRAS mutants, sustaining their oncogenic activity. Conversely, the enhanced recruitment of LAMTOR1, a lysosomal scaffold essential for mTORC1 activation, was observed in all KRAS G12C/D/V mutants. These opposing changes (LZTR1 loss and LAMTOR1 gain) collectively sustain KRAS activation while rewiring metabolic networks. LZTR1, functioning as an adaptor for the Cullin3 (CUL3) ubiquitin ligase complex, mediates polyubiquitination-mediated proteasomal degradation of RAS-GTPases, including KRAS, through K48-, K63-, and K33-linked ubiquitin chains. This process suppresses aberrant RAS/MAPK signaling activation. Notably, LZTR1 retains its regulatory capacity over oncogenic KRAS mutants (e.g., G12V/D/C), suggesting its potential as a broad-spectrum RAS suppressor. While LZTR1 interacts with autophagy-related proteins like LC3B and SQSTM1/p62, its degradation of KRAS primarily relies on the ubiquitin-proteasome system, though its role in autophagy appears minimal. Clinically, LZTR1 expression is frequently downregulated in KRAS-driven malignancies such as pancreatic and colorectal cancers, correlating with elevated KRAS stability and MAPK hyperactivation. Furthermore, LZTR1 loss-of-function mutations linked to Noonan syndrome and schwannomatosis underscore its critical role in maintaining RAS signaling homeostasis, providing a molecular rationale for therapeutic strategies targeting KRAS degradation [43]. LAMTOR1’s role in lysosomal signaling positions it as a promising therapeutic node. Recent studies demonstrate that modulating LAMTOR1 enhances immunotherapy efficacy in NSCLC by reducing exosomal PD-L1 [44] and synergizes with chemotherapy via cGAS-STING activation [28]. Additionally, the TRAF4-mediated ubiquitination of LAMTOR1 drives mTORC1 activation in colorectal cancer, suggesting that TRAF4 inhibitors could suppress oncogenic signaling [45]. Notably, vitamin E analogs inhibit LAMTOR1-HDAC6 interactions to attenuate NLRP3-driven inflammation [46]. While our study focuses on LAMTOR1’s interactome role, these findings collectively propose targeting its ubiquitination (K151) or lysosomal binding domains as viable strategies, pending validation in KRAS-mutant models.

While our proteomic framework clarifies mutation-specific mechanisms, several limitations warrant consideration to guide future investigations. First, the reliance on HEK293T overexpression systems may not fully recapitulate native tumor microenvironment dynamics, necessitating validation in patient-derived organoids or genetically engineered mouse models (GEMMs) to enhance clinical relevance. Second, although TurboID enables efficient proximity labeling, prolonged labeling periods risk the bystander biotinylation of non-proximal proteins, potentially confounding interaction specificity. Third, while LZTR1 and LAMTOR1 emerge as key regulatory nodes across KRAS G12 mutants, their mechanistic roles in metabolic reprogramming require direct interrogation through CRISPR-based functional studies in disease-relevant contexts. Fourth, the clinical significance of LAMTOR1 overexpression observed in TCGA cohorts demands prospective validation using matched pre-/post-treatment samples from KRAS inhibitor trials. Fifth, our focus on KRAS G12C/D/V mutants leaves other clinically relevant variants (e.g., KRAS G12R/S/A) unexplored, limiting a comprehensive understanding of KRAS mutational spectra. Finally, the therapeutic potential of targeting identified interactors remains to be tested in pharmacologically relevant systems, particularly given tissue-specific mutation prevalence. Addressing these limitations—spanning model systems, technical specificity, functional validation, and clinical translation—will be critical for advancing these findings toward actionable therapies.

## 5. Conclusions

In summary, our findings revealed that multiple intracellular metabolic pathways respond to KRAS G12 mutations, with distinct metabolic remodeling observed across different mutation types. Notably, the three G12 mutants exhibited shared alterations in the molecular environment of KRAS, with key regulatory proteins such as LZTR1 and LAMTOR1. These findings not only deepen our understanding of KRAS-driven malignant proliferation and carcinogenesis but also provide new directions for future therapeutic development by targeting KRAS mutants associated signaling pathways.

## Figures and Tables

**Figure 1 biology-14-00477-f001:**
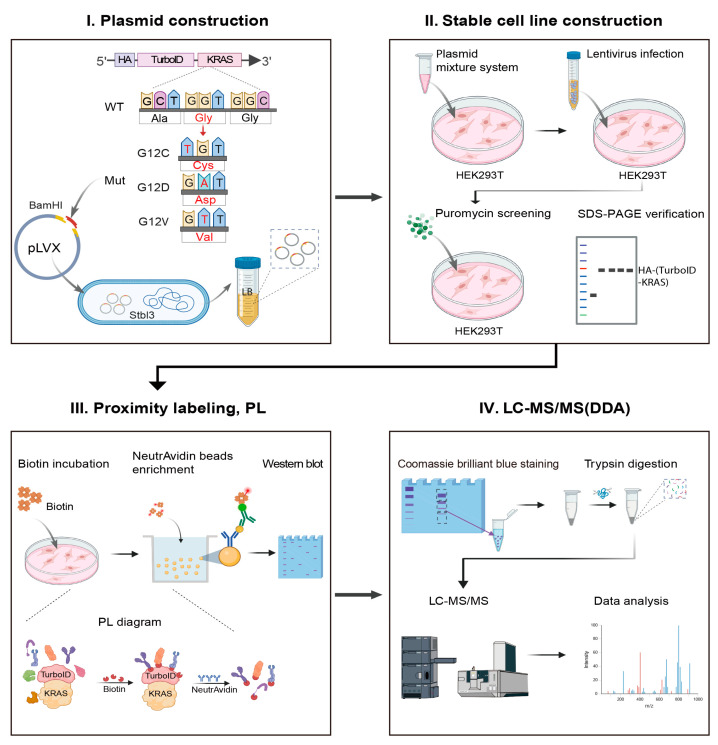
Schematic workflow of this study.

**Figure 2 biology-14-00477-f002:**
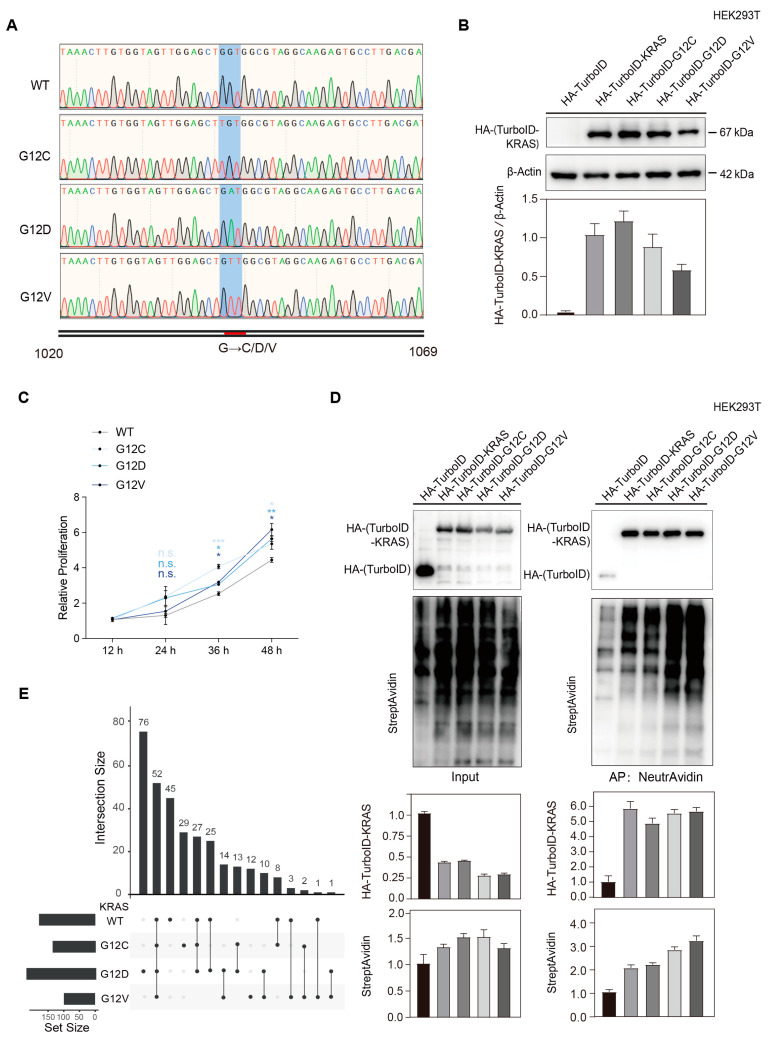
Establishment of KRAS variant cell models and proximity labeling of KRAS-interacting proteins. (**A**) DNA sequencing profile confirms the correct assembly of KRAS WT and G12C/D/V mutant plasmids. (**B**) Western blot analysis verifies the stable expression of HA-TurboID-KRAS fusion proteins in respective cell lines. Bar chart below shows the relative dose of proteins analyzed by gray analysis. (**C**) CCK-8 assay quantifies the proliferation kinetics of WT versus G12 mutant cell lines. Data are presented as mean ± SD. *p* values were determined by one-way ANOVA, with light to dark blue shades representing comparisons between KRAS G12C/D/V and KRAS WT groups, respectively. Statistical significance is denoted as follows: * *p* < 0.05, ** *p* < 0.01, *** *p* < 0.001. (**D**) Streptavidin blot reveals the specific biotinylation of KRAS proximal proteins in experimental groups. Bar charts below show the relative dose of proteins analyzed by gray analysis. (**E**) UpSet plot shows a comparative analysis of interactome overlaps among WT and G12 mutants.

**Figure 3 biology-14-00477-f003:**
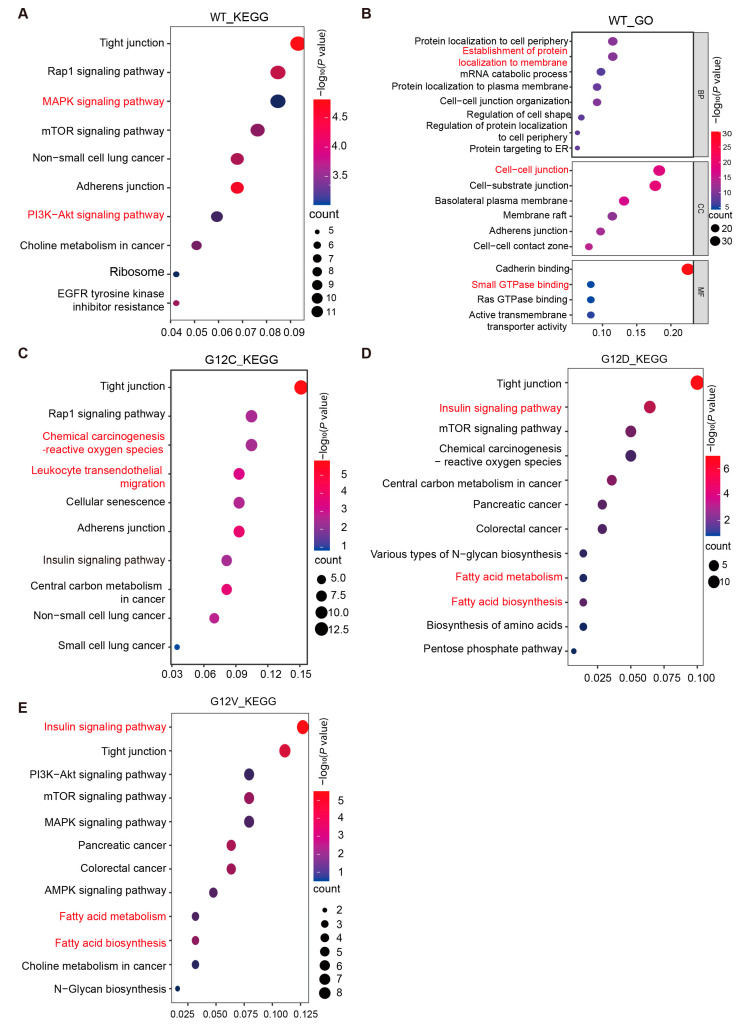
Enrichment analysis of KRAS WT- and G12C/D/V-interacting proteins. (**A**,**B**) KEGG and GO analysis of KRAS WT-interacting proteins. (**C**–**E**) KEGG analysis of G12C (**C**), G12D (**D**), and G12V (**E**)—interacting proteins, respectively.

**Figure 4 biology-14-00477-f004:**
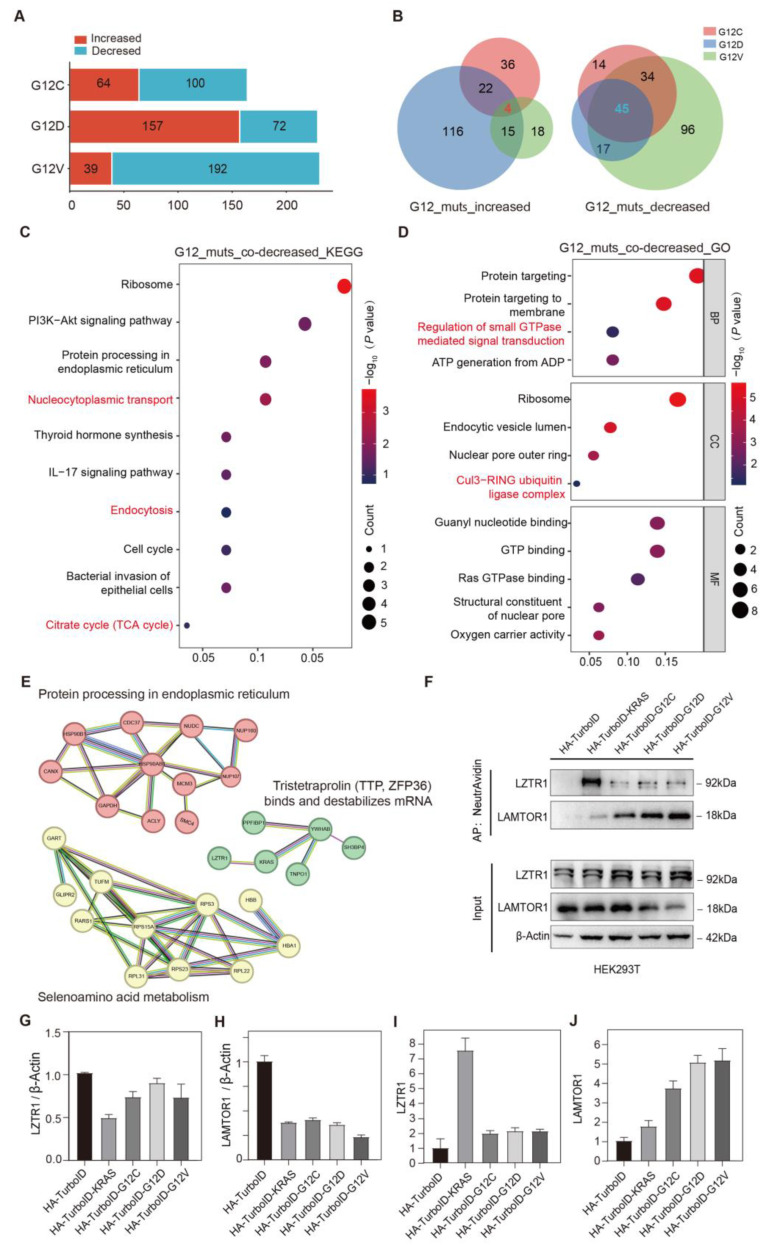
Pathway enrichment analysis of simultaneously changed interacting proteins in all G12C/D/V mutant groups. (**A**) Bar chart showing the number of increased and decreased binding proteins in the G12C, G12D, and G12V groups compared to the WT group. (**B**) Venn diagram showing the increased and decreased binding proteins of the G12 mutant groups. Light red represents the G12C group, light blue indicates the G12D group, and light green denotes the G12V group. (**C**,**D**) KEGG (**C**) and GO (**D**) annotation of the co-decreased binding proteins in G12C, G12D, and G12V groups. (**E**) PPI analysis of the co-decreased binding proteins in G12C, G12D, and G12V groups. (**F**–**J**) Western blot validation and gray analysis of representative co-decreased binding protein LZTR1 and co-increased binding protein LAMTOR1.

**Table 1 biology-14-00477-t001:** Primers of HA-TurboID-KRAS construction.

Type	Sequence (5′-3′)
Forward 1	AAAGACAATACTGTGCCTCTGAA
Reverse 1	GTTTATATTCAGTCATCTTTTCGGCAGACCGCA
Forward 2	ATGACTGAATATAAACTTGTGGTAGTTGG
Reverse 2	TTATCTAGAGTCGCGGGATCCTTACATTATAATGCATTTTTTAATTTTCACACAG

**Table 2 biology-14-00477-t002:** Primers for the construction of HA-TurboID-KRAS-G12 mutants.

Subtype	Mutated Base	Sequence (5′-3′)
G12C	GGT-TGT	F: TTGGAGCTTGTGGCGTAGGCAAGAGTGCCTTG
		R: TACGCCACAAGCTCCAACTACCACAAGTTTTATTCA
G12D	GGT-GAT	F: TTGGAGCTGATGGCGTAGGCAAGAGTGCCTTG
		R: TACGCCATCAGCTCCAACTACCACAAGTTTATATT
G12V	GGT-GTT	F: TTGGAGCTGTTGGCGTAGGCAAGAGTGCCTTG
		R: TACGCCAACAGCTCCAACTACCACAAGTTTATATT

## Data Availability

The data that support the findings of this study are available from the corresponding author upon reasonable request. Mass spectrometry proteomics data have been deposited into the Proteome Xchange Consortium via the *PRIDE* [43] partner repository with dataset identifier PXD062101. Reviewer account details: username: reviewer_pxd062101@ebi.ac.uk; password: dXOXja8GXJkS.

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
