# Peer review of "Mapping the Interactome of KRAS and Its G12C/D/V Mutants by Integrating TurboID Proximity Labeling with Quantitative Proteomics"

_biology, 2025, doi:10.3390/biology14050477_

Round 1
Reviewer 1 Report
Comments and Suggestions for Authors
In this manuscript, Jiangwei Song et al. present a comprehensive approach to dissecting the mutation-specific interactome of KRAS G12 variants in comparison to wild-type (WT) KRAS. The integration of TurboID-based proximity labeling with quantitative proteomics is well-executed, and the subsequent bioinformatics analyses yield meaningful biological insights, particularly in the context of metabolic rewiring and potential therapeutic targeting. The manuscript is well-structured, clearly written, and should be of broad interest to the Biology readers
Comments:
1. In Figure 3A, the KRAS-WT interactome is described as being associated with MAPK and PI3K-AKT signaling. However, based on the p-values, these associations appear relatively weaker compared to other pathways highlighted in the figure. The authors may wish to rephrase the related text for clarity and accuracy.
2. In Figure 4B, the color coding in the Venn diagram is unclear. Including a legend or adding details to the figure would improve clarity.
3. In Figure 4F (G12V group), despite relatively low LAMTOR1 expression in the input sample, the protein shows the highest enrichment in the NeutryAvidin affinity purification. Could the authors provide an explanation for this discrepancy?
4. The observed reduced interaction with LZTR1 is interesting, but its impact on KRAS stability remains unexplored. The authors could consider experiments involving KRAS overexpression or knockdown to test this directly.
5. The identification of LAMTOR1 as a significant interactor is intriguing; however, no direct evidence is presented linking its modulation to therapeutic outcomes. A discussion of potential therapeutic strategies or available drugs that could target this node would enhance the impact of the findings.
6. Lines 250–254: Please consider adding relevant citations to support the validated findings referenced in this section.
7. Line 294 contains a typographical error—"C" is missing after "G12". If referring to the G12C mutant, please correct accordingly.
8. Several figures suffer from low resolution and text blurriness, making them difficult to read. Improving figure quality is recommended.
9. Consider adding a paragraph explicitly discussing the limitations of the study, especially the need for validation in cancer-relevant model systems.
10. While the methods are rigorous, some results (e.g., Figure 2C proliferation data) lack p-values and error bars. Please ensure statistical details are provided where appropriate.
11. The manuscript alternates between the terms “G12 mutants” and “G12C/D/V”. For consistency, consider consistent terminology throughout (e.g., “KRAS G12C/D/V mutants” or “G12-mutants”).
12. Line 198: The sentence reads, “The data are presented as the mean ± SD taken from no fewer than three independent experiments.” Please consider specifying the exact number of replicates (e.g., “n = 3”) for clarity.
Reviewer 2 Report
Comments and Suggestions for Authors`
The manuscript titled "Mapping the Interactome of KRAS and Its G12C/D/V Mutants by Integrating TurboID Proximity Labeling with Quantitative Proteomics" utilizes TurboID proximity labeling linked to quantitative proteomics to investigate the influence of KRAS and 3 different KRAS mutants in tumor progression. Their study discovers LZTR1 and LAMTORC1 as plausible therapeutic candidates for KRAS-driven malignancy.
Major Concern:
Figure quality in the main text is very poor. The letters in the figure are harder to read and follow because of the clarity. The authors should provide good quality images for the journal to publish.
Minor Concerns:
In Figure 3 and 4, in the GO and KEGG plots, what do those letters in red indicate? Who are those pathways chosen to be discussed than the other top candidates from the plot?
Line 333: Did the authors do further validation on LZTR1 and LAMTOR1 to show that their influence can be modulated in KRAS mutations?
Also, did the authors consider doing any phenotypic validation studies based on their pathway analyses outcomes?
